# Potential Biomarkers for Predicting Congenital Cytomegalovirus Infection

**DOI:** 10.3390/ijms19123760

**Published:** 2018-11-27

**Authors:** Kenji Tanimura, Hideto Yamada

**Affiliations:** Department of Obstetrics and Gynecology, Kobe University Graduate School of Medicine, Kobe 650-0017, Japan; kobeobgy@med.kobe-u.ac.jp

**Keywords:** biomarker, congenital infection, cytomegalovirus, prediction, screening

## Abstract

Early diagnosis and treatment of infants with symptomatic congenital cytomegalovirus (CMV) infection may improve neurological outcomes. For this reason, prenatal detection of newborns at high risk for congenital CMV infection is important. A polymerase chain reaction (PCR) assay for CMV DNA in the amniotic fluid is the gold standard for the diagnosis of intrauterine CMV infection; however, amniocentesis is an invasive procedure. Recently, we have found that the presence of CMV DNA in the maternal uterine cervical secretion is predictive of the occurrence of congenital CMV infection in CMV immunoglobulin M (IgM)-positive pregnant women. In contrast, we have suggested that maternal serological screening for primary CMV infection using CMV-specific immunoglobulin G (IgG), the IgG avidity index, or CMV-specific IgM overlooks a number of newborns with congenital CMV infection. We will review current knowledge of the potential biomarkers for predicting congenital CMV infection.

## 1. Introduction

Cytomegalovirus (CMV) is a common cause of congenital infection. The incidence of congenital CMV infection is reported to be 0.2–2.4% in infants in developed countries [1], and 10–15% of affected fetuses show symptomatic congenital infection at birth. The clinical manifestations of congenital CMV infection, including fetal growth restriction (FGR), low birth weight, and cerebral and multiple organ involvement, can be so severe that they cause major neurological sequelae in about 90% of surviving infants. Additionally, 10–15% of newborns with asymptomatic congenital CMV infection develop long-term sequelae—for example, progressive sensorineural hearing difficulty and psychomotor retardation [2]. For example, in the United States, an estimated 40,000 children with congenital CMV infection are born annually, resulting in an estimated 8000 children who have long-term sequelae. In recent years, the annual economic costs spent on the care of affected children are estimated at more than 3 billion dollars [3].

Early diagnosis and early intervention with antiviral drugs have been shown to improve neurological outcomes in infants with symptomatic congenital CMV infection [4,5]. Therefore, prenatal detection of high-risk pregnancies with congenital infection is critical for accurately diagnosing congenital CMV infection, and for early treatment for symptomatic newborns.

On the other hand, preemptive medicine is a new preventive strategy in the medical field. In this strategy, clinicians predict the risk of diseases and complications in individual patients by the use of biomarkers, and they intervene in the high-risk patients to delay or prevent the onset of diseases. If we can develop preemptive strategies for congenital CMV infection, the health and economic burden of the disease can be reduced.

However, neither maternal nor neonatal universal screening for congenital CMV infection has been recommended worldwide, because vaccines and established fetal or neonatal treatments are not yet available.

This review focuses on current knowledge of the potential biomarkers that may lead to the development of preemptive strategies for congenital CMV infection and disease.

## 2. Cytomegalovirus-Specific Antibody Tests

Not only primary but also non-primary infection (reactivation or reinfection with a different CMV strain) can cause congenital infection, although preexisting maternal immunity against CMV is related to a 69% decrease in the risk of congenital infection [6]. Approximately 40% of fetuses whose mothers have primary CMV infection during pregnancy will be infected [6]. Therefore, maternal serological tests for detecting pregnant women with primary CMV infection, including maternal blood tests of CMV-specific immunoglobulin G (CMV IgG) and CMV-specific immunoglobulin M (CMV IgM) have been widely used. The gold standard for the diagnosis of primary CMV infection is the detection of CMV IgG seroconversion. In many pregnant women who were positive for CMV IgG during pregnancy, however, it was unknown whether they were negative for CMV IgG just before or after ongoing pregnancy. Therefore, tests for maternal serum CMV IgM are commonly used to identify primary CMV infection during or just before ongoing pregnancy. However, positive CMV IgM yielded 20–25% sensitivity [7] and a 15–20% false-positive rate [8] for detecting primary CMV infection, because CMV IgM may persist for 6–9 months following primary CMV infection [9] or may be detected during latent CMV reactivation [7].

## 3. Cytomegalovirus Immunoglobulin G Avidity Measurements

The serum CMV IgG avidity index is used as a confirmatory test to identify primary CMV infection [10]. The CMV IgG avidity index is measured by co-incubating the serum with or without 6 M urea as a dissociating agent in an immunoenzymatic assay. Because the CMV IgG avidity index increases over time, low and high CMV IgG avidity indices indicate recent and previous CMV infections, respectively. Previous studies have demonstrated that a low CMV IgG avidity index is a significant predictor of congenital CMV infection [11,12].

In a prospective cohort study, 759 pregnant women who were positive for CMV IgG received CMV IgG avidity measurements (Aisenkai Nichinan Hospital, Miyazaki, Japan), and congenital CMV infection was confirmed by polymerase chain reaction (PCR) assays for CMV DNA in newborn urine [12]. A cutoff value of the CMV IgG avidity index for predicting the occurrence of congenital CMV infection was determined using receiver operating characteristic (ROC) analysis. Fourteen (1.8%) of the 759 pregnant women had congenital CMV infection, and the ROC analysis had revealed that a cutoff value of <40% of the CMV IgG avidity index yielded the best results, with 64.3% sensitivity and 96.1% specificity for the prediction of congenital CMV infection. Specifically, a cutoff value of <40% CMV IgG avidity index in pregnant women who underwent CMV IgG avidity measurements before 28 weeks of gestation (GW) yielded the best results, with sensitivity, specificity, positive predictive value (PPV), and negative predictive value (NPV) of 88.9%, 96.2%, 27.6%, and 99.8%, respectively, for the prediction of congenital CMV infection [12].

In addition, we have reported that the rate of change in the CMV IgG avidity index per 4 weeks, which was defined as the “Δ avidity index”, of >10% yielded 100% PPV for predicting congenital CMV infection among pregnant women with positive or equivocal tests for CMV IgM, together with an initial CMV IgG avidity index of <40% [13]. These results indicated that a rapid increase in the serum CMV IgG avidity index was associated with the occurrence of congenital CMV infection among pregnant women with suspected primary CMV infection during pregnancy.

In contrast, in a prospective cohort study of 2193 low-risk pregnant women (i.e., pregnant women not referred because of positive results for CMV IgM), the prediction accuracy for congenital CMV infection of maternal serological screening, based on CMV IgG avidity index (cutoff values: ≤45% or < 35%), was similar to that based on CMV IgM (>1.2 index) [14]. In addition, in this study, 10 of 2193 (0.5%) pregnant women delivered newborns with congenital CMV infection, and 3 of 10 infants with congenital CMV infection were delivered to pregnant women who had primary CMV infection during pregnancy, defined as a CMV IgG avidity index of < 35%, CMV IgM >1.2 index, or detection of CMV IgG seroconversion. Surprisingly, the remaining seven congenitally infected infants were delivered to pregnant women with previous CMV infection, defined as a CMV IgG avidity index >45% and CMV IgM ≤1.2 index. These results suggest that universal screening using CMV IgG avidity measurements in low-risk pregnant women is inefficient for the prediction of congenital infection, because the majority of congenital CMV infection was caused by non-primary CMV infection (i.e., reactivation or reinfection with a different strain). Recently, many studies have also demonstrated that the major of congenital CMV infection may be caused by non-primary CMV infection, and that the prevalence of symptomatic congenital CMV infection in infants born to mothers with non-primary infection is similar to that in infants born to pregnant women with primary infection [15]. Therefore, a high CMV IgG avidity index cannot be considered a source of reassurance.

## 4. Epitope-Specific Antibody Detection

Human CMV contains multiple surface-expressed glycoproteins that play pivotal roles in viral entry, including glycoprotein B (gB), the complex of glycoprotein M (gM) with glycoprotein N (gN) (gM/gN), the complex of glycoprotein H (gH) with glycoprotein L (gL) (gH/gL), and a pentameric complex of gH/gL/unique long (UL)128/UL130/UL131A. In particular, the pentameric complex is responsible for viral entry into endothelial, epithelial, monocytic, and dendritic cells, and the complex decides viral tropism for diverse cells [16,17]. Recent studies indicate that the pentameric complex induces not only higher levels but also higher binding avidities of neutralizing antibodies than gB or the gH/gL; therefore, the pentameric complex is a potential candidate for human CMV vaccines [18,19,20]. In addition, it has been demonstrated that a delay in the production of maternal antibodies against the pentameric complex during primary infection is associated with mother-to-fetus CMV transmission [21], and that both antibodies against the pentameric complex and higher CMV IgG avidity index are correlated with a decreased risk of congenital infection [22]. Therefore, the pentameric complex is thought to be a major target for the development of vaccines to prevent fetal CMV infection.

On the other hand, because the neutralizing antibody responses appear after an average of 13 weeks after CMV IgG seroconversion, the absence of neutralizing antibodies in CMV IgG positive-pregnant women is thought to be a useful biomarker for primary CMV infection. The previous study reported that an immunoblot assay for detecting neutralizing antibodies against epitopes of glycoproteins (gp)—e.g., glycoprotein B (gB) (gpUL55) and glycoprotein H (gH) (gpUL75) of human CMV—yielded 93.6% sensitivity and 100% specificity for identifying previous CMV infection [23]. Furthermore, this study also suggested that delayed appearance of gB-specific antibodies in primary CMV infection was associated with a lower risk of congenital CMV infection.

On the other hand, the previous studies suggested that pregnant women who had reinfection with different strains of CMV were at higher risk of congenital CMV infection than those who had CMV reactivation during pregnancy [24]. Novak et al. developed an enzyme-linked immunosorbent assay (ELISA) method to identify serological responses against reinfection with different strains of CMV, based on the defined heterogeneity in the antibody binding epitopes on envelope glycoproteins gB and gH of the laboratory strains of CMV (AD169 and Towne strains) [25]. In this study, serum levels of strain-specific antibodies were measured in 96 seropositive women by this ELISA, and 58 (60%) women were positive for at least one autoantibody of the antibodies against the four antigens, and 18 women were positive for two or more antibodies. These results indicated that this ELISA method may be useful for identifying reinfection with different strains. However, this assay couldn’t identify all reinfection, because there may be other polymorphic epitopes on gH and gB, or there may be other envelope glycoproteins of CMV (for example, gN). In addition, these assays for detecting epitope-specific antibodies are not commonly available, because they are non-standardized in-house assays, and they requires complex data interpretation.

## 5. Cytomegalovirus DNA Polymerase Chain Reaction Assay

Previous studies have found that a CMV DNA PCR assay of the amniotic fluid, including nested, single-round, or real-time PCR, yielded 75–100% sensitivity and 67–100% specificity for prenatal diagnosis of fetal CMV infection [26,27]. The PCR assay is the gold standard for prenatal diagnosis of fetal CMV infection, because CMV DNA PCR tests of the amniotic fluid are a highly sensitive and specific method for identifying fetal infection. In contrast, CMV is excreted into the amniotic fluid through fetal urine; therefore, amniocentesis should be performed after 20–21 GW, when fetal urination is established. In addition, because the CMV viral load in the amniotic fluid does not achieve detectable levels until 6–9 weeks after maternal CMV infection, amniocentesis followed by CMV DNA PCR analysis should be performed at least 6 weeks after primary maternal CMV infection and after 21 GW, to reduce false-negative results [28].

However, because amniocentesis is essentially an invasive procedure with a risk of rupture of the membranes, it is not realistic that the amniotic fluid of all pregnant women suspected with primary CMV infection should be analyzed using CMV DNA PCR. Recently, we conducted a prospective cohort study to determine maternal clinical, laboratory, and imaging findings that can effectively and noninvasively predict the occurrence of congenital CMV infection among pregnant women who were positive for CMV IgM [29]. In this prospective cohort study of 300 pregnant women positive for CMV IgM, including 22 pregnant women who delivered newborns with congenital CMV infection, maternal serum CMV IgG avidity index; an antigenemia (C7-HRP) assay (CMV antigen test); CMV DNA PCR assays in the maternal serum, urine, and maternal uterine cervical secretion; and fetal ultrasound findings were evaluated. By a stepwise approach, using univariate and multivariable logistic regression analyses, it was revealed that positive results for CMV DNA PCR in the maternal uterine cervical secretion (odds ratio [OR], 16.4; 95% confidence interval [CI], 5.0–54.1; *p* < 0.001), and the detection of ultrasound fetal abnormalities (OR, 31.9; 95% CI, 8.5–120.3; *p* < 0.001) were significant predictors for congenital CMV infection in CMV IgM-positive pregnant women. The positive CMV DNA PCR results in the uterine cervical secretion yielded sensitivity, specificity, PPV, and NPV of 50.0%, 94.2%, 40.7%, and 96.0%, respectively, for the prediction of congenital CMV infection. We proposed three hypotheses for the association between the detection of CMV DNA in uterine cervical secretion and the occurrence of congenital CMV infection. First, CMV is transmitted to the fetus via the genital tract by ascending infection. Second, CMV shedding in the uterine cervical secretion persists after maternal primary infection or reinfection with a different strain, both of which cause fetal CMV infection. Third, CMV DNA in the amniotic fluid leaks into the genital tract.

On the other hand, in this study, the proportion of pregnant women positive for CMV DNA PCR in the maternal urine was not different between a group of pregnant women with congenital infection (*n* = 22) and those without congenital infection (*n* = 278) (14% versus 5%; *p* = 0.2). Furthermore, there were no pregnant women positive for CMV DNA PCR in the serum during the study period. These results suggest that neither PCR assays for CMV DNA in the maternal urine nor those in the maternal serum are useful for predicting the occurrence of congenital CMV infection.

In addition, our other previous studies have indicated that the prediction of congenital CMV infection by PCR assays for CMV DNA in the uterine cervical secretion may be inefficient when enrollment of subjects were not limited to CMV IgM-positive pregnant women [14].

## 6. Assays for Measuring Cytomegalovirus-Specific, T Cell-Mediated Immunity

Many studies have suggested that CMV-specific, T cell-mediated immune responses play a crucial role in controlling viral replication and severity of disease; however, the specific features of the responses that contribute to protection against fetal infection remain unclear [30].

Previous studies have demonstrated that phosphoprotein (pp)65-specific cluster of differentiation 8-positive (CD8+) and immediate–early antigen (IE1)-specific CD8+ T cells have a protective function against CMV viremia in transplant recipients [31,32], and that pp65-specific CD4+ T cells may play a crucial role in the protection against mother-to-fetus CMV transmission [33].

In contrast, recent studies have demonstrated that strong, CMV-specific, T cell-mediated immunity is associated with the occurrence of congenital CMV infection [34,35]. Interferon-γ release assays, including enzyme-linked immunosorbent spot (ELISPOT) and QuantiFERON (QFT) assays, are widely used to evaluate the T cell-mediated immunities of patients. A previous study of 80 pregnant women with possible active CMV infection had revealed that pregnant women with primary CMV infection had significantly higher CMV-specific, T cell-mediated immune responses compared with those with non-primary CMV infection, including reactivation or reinfection with a different strain. Moreover, the study had also revealed that the maternal CMV-specific, T cell-mediated immunity in pregnant women with primary CMV infection was positively correlated with the incidence of congenital CMV transmission. In particular, pregnant women with a CMV-specific T cell response of >185 spots /2 × 10^5^ peripheral blood mononuclear cells were found to be at high risk for congenital CMV infection, regardless of primary or non-primary infection [34].

The author of the study speculated that high cell-mediated immune responses may promote CMV transmission in primary infection, whereas the preexisting cell-mediated immunity in non-primary infection may exert protective effects against fetal infection [34].

In addition, the CMV ELISPOT assay, not the CMV QFT assay, has been reported to discriminate between pregnancies with mother-to-fetus CMV transmission and those without CMV transmission [35,36].

Thus, the results of assays for measuring CMV-specific, T cell-mediated immunity are sometimes difficult to interpret.

## 7. Imaging Examinations

Unless maternal serological CMV screening is performed, the presence of ultrasound fetal abnormalities associated with congenital CMV infection during the second or third trimester is the test result that motivates clinicians to suspect that the fetus has congenital CMV infection. Ultrasound fetal abnormalities, including ventriculomegaly (4.5–11.6%), microcephaly (14.5%), intracranial calcification (0.6–17.4%), FGR (1.9–13%), pericardial effusion (7.2%), ascites (8.7%), hepatomegaly (4.3%), and intestinal high echodensities (4.5–13%), are known to be predictive of symptomatic congenital CMV infection [11,37,38,39]

As described previously, our prospective cohort study had demonstrated that the presence of ultrasound fetal abnormalities was one of the most significant predictive factors for congenital CMV infection among CMV IgM-positive pregnant women [29]. In this prospective study, the presence of ultrasound fetal abnormalities yielded sensitivity, specificity, PPV, and NPV of 50.0%, 97.5%, 61.1%, and 96.1%, respectively, for the prediction of congenital infection. Of the 300 pregnant women who were positive for CMV IgM, 18 (6%) had at least one ultrasound finding that was suggestive of congenital CMV infection. Of the 18 pregnant women with ultrasound fetal abnormalities, 11 were confirmed to have congenital CMV infection. We found that ultrasound findings of ventriculomegaly, intracranial calcification, microcephaly, and hepatosplenomegaly were specific to congenital CMV infection, and that those of FGR, pleural effusion or ascites, and hyperechoic bowel were not specific [29].

In addition, some investigators suggested that only severe cranial abnormalities, such as ventriculomegaly and microcephaly, were associated with poor prognosis in surviving infants with symptomatic congenital CMV infection [37].

In contrast, magnetic resonance imaging (MRI) is thought to be more sensitive in detecting fetal intracranial abnormalities associated with congenital CMV infection than ultrasound examinations [40]. However, neuroradiologists should evaluate MRI images, because the results are often difficult to interpret.

## 8. Clinical Factors Associated with the Occurrence of Congenital Cytomegalovirus Infection

Our prospective cohort study had indicated that maternal universal screening for congenital CMV infection based on serological tests—including CMV IgG, IgG avidity index, and CMV IgM—in low-risk pregnant women (i.e., pregnant women who were not referred because of positive results for CMV IgM) overlooked a number of congenitally infected newborns, because a larger number of congenital CMV infections were caused by maternal non-primary than by primary CMV infection [14]. In addition, recent studies have demonstrated that the number and severity of symptoms in infants with congenital CMV infection from pregnant women with non-primary CMV infection were not inferior to those from pregnant women with primary CMV infection [41,42]. Currently, the prediction of congenital CMV infection from pregnant women with non-primary CMV infection by using laboratory or biological tests is not yet possible [43]. Therefore, by a nested case–control study, we evaluated clinical factors related to the occurrence of congenital infection among pregnant women with non-primary CMV infection [44]. This study enrolled 1287 pregnant women with previous CMV infection, which was defined as CMV IgM < 1.2 index and CMV IgG avidity index >45%, and seven newborns (0.5%) had congenital infection. Univariate logistic regression analyses revealed that multiple pregnancy (OR, 7.1; 95% CI 1.4–37.4; *p* < 0.05) and threatened premature delivery (OR, 10.6; 95% CI, 2.0–55.0; *p* < 0.01) were associated with the occurrence of congenital CMV infection. In addition, univariate logistic regression analyses also indicated that maternal fever or flu-like symptoms (OR, 3.7; 95% CI, 0.8–16.9; *p* = 0.09) and preterm delivery (OR, 4.0; 95% CI, 0.9–18.1; *p* = 0.07) tended to be associated with the occurrence of congenital CMV infection. Multivariable logistic regression analyses revealed that threatened premature delivery (OR, 8.4; 95% CI, 1.5–48.1; *p* < 0.05) was an independent risk factor for congenital infection in pregnant women who had previous CMV infection [44]. We proposed two hypotheses on the association between threatened premature delivery and occurrence of congenital infection of CMV in pregnant women who have non-primary CMV infection. First, it is possible that intrauterine CMV infection causes threatened premature delivery. Second, inflammatory conditions underlying threatened premature delivery perhaps reactivate latent CMV via cytokine induction, and lead to mother-to-fetus CMV transmission.

## 9. Neonatal Screening and Definitive Diagnosis of Congenital Cytomegalovirus Infection

It remains controversial whether neonatal universal screening or targeted screening for congenital CMV infection should be performed.

Regarding neonatal CMV screening, some investigators universally used PCR assays for CMV DNA from the saliva or urine of newborn [5,45]. In neonatal screening based on CMV DNA PCR assays from the newborn’s saliva, CMV shedding into the maternal breast milk may cause false-positive results [46]; therefore infants who had positive PCR results in the saliva should receive confirmatory testing by CMV DNA PCR assay from the urine.

On the other hand, because dried blood spots are collected routinely for newborn metabolic screening from all infants born in the United States, Japan, and so on, there has been considerable interest in neonatal universal screening for congenital CMV infection based on PCR assay from the newborn dried blood spot samples. However, in the largest prospective study, 20,448 newborns were screened for congenital CMV infection by real-time PCR analysis of dried blood spots. This study demonstrated that the screening for congenital CMV infection based on the dried blood spots PCR tests yielded 28.3% sensitivity and 99.9% specificity [47]. The same study group also reported that the screening for congenital infection using the PCR assays from the liquid saliva yielded 100% sensitivity and 99.9% specificity [48].

In contrast, in neonatal targeted CMV screening, infants who failed neonatal hearing screening and referred for audiological testing receive a confirmatory test based on PCR assays for CMV DNA from the urine [49]. However, neonatal targeted CMV screening may overlook a number of infants with congenital CMV infection, because most of them are asymptomatic and can have delayed onset of CMV-related hearing loss [49].

Conversely, in a prospective multicenter clinical trial, 6348 infants underwent universal screening based on PCR assays for CMV DNA in the urine [5]. Thirty-two infants (0.5%) were positive for CMV DNA in the urine, and 16 (50%) of the 32 were diagnosed with symptomatic congenital CMV infection. Twelve of the 16 infants with symptomatic congenital CMV infection received oral valganciclovir (16–32 mg/kg/day) for six weeks. In the 16 infants who received antiviral treatments, four infants (33%) showed severe impairment, three (25%) showed mild impairment, and five (42%) showed normal development at a median of 37 months of age. Previous reports have estimated that 70–90% of infants with symptomatic congenital CMV infection had severe sequelae [50]; therefore these results indicate that the combination of early detection by universal neonatal screening based on PCR assays for CMV DNA in the urine and early antiviral treatments may improve outcomes in infants with symptomatic congenital CMV infection [5].

## 10. Potential Strategies to Improve the Outcomes of Children with Congenital CMV Infection

Figure 1 is a flow algorithm, which shows our recommendation for the diagnosis and treatment of congenital CMV infection. The CMV IgG levels in pregnant women are measured in the first trimester. Pregnant women who are negative for CMV IgG receive educational intervention for preventing primary CMV infection during their ongoing pregnancies, and CMV IgG levels are measured again from 34 to 36 GW. On the other hand, CMV IgG seroconversion among CMV IgG-negative women is regarded as primary infection during pregnancy. Pregnant women positive for CMV IgG are tested for serum CMV IgM, and pregnant women positive for CMV IgM receive CMV IgG avidity measurement. Pregnant women who are negative for CMV IgM, or those with a high CMV IgG avidity index, are considered to have had previous CMV infection (in other words, latently CMV infection) during pregnancy. Conversely, pregnant women with a low CMV IgG avidity index together with a positive result for CMV IgM are considered to have primary CMV infection during pregnancy. If pregnant women with primary CMV infection desire further examinations, the amniocentesis followed by CMV DNA PCR analysis of the amniotic fluid is performed with informed consent. Although there is no established fetal therapy for symptomatic congenital CMV infection, fetal therapies, e.g., hyper-immunoglobulin injection into maternal blood [51], into the fetal peritoneal cavity [52], or oral administration of high-dosage valacyclovir to mothers [53], etc., might be considered for fetuses diagnosed with symptomatic congenital CMV infection (i.e., their mothers have both positive CMV DNA PCR results from the amniotic fluid and ultrasound fetal abnormalities related to congenital CMV infection). In contrast, not only pregnant women with primary CMV infection, but also those with previous CMV infection should undergo periodic ultrasound examinations to detect fetal abnormalities associated with congenital CMV infection throughout pregnancy.

Infants of mothers with primary CMV infection during pregnancy or those with fetal ultrasound abnormalities indicative of congenital CMV infection, should undergo PCR for CMV DNA in newborn urine to confirm the presence of congenital CMV infection. Newborns with positive CMV DNA PCR results from the urine should be screened to identify the symptoms of congenital CMV infection, including ophthalmoscopy, cerebral ultrasound, physical and neurological examinations, head computed tomography, head MRI, and auditory brainstem response test. There is no established consensus regarding the efficacy of neonatal treatments for symptomatic congenital CMV infection. However, because recent studies have suggested that early intervention with antiviral drugs may improve neurological outcomes in infants with symptomatic congenital CMV infection, neonatal therapies, e.g., intravenous administration of ganciclovir [4] or oral administration of valganciclovir [5], might be considered.

## 11. Conclusions

Recently, it has been reported that the pentameric complex, i.e., the gH/gL/UL128/UL130/UL131A complex, is a potential candidate for human CMV vaccines [18,19]. Furthermore, the pentameric complex is thought to be a major target for the development of vaccines to prevent fetal CMV infection. On the other hand, albeit in a preclinical stage, new vaccines (e.g., enveloped virus-like particles vaccines, vectored CMV vaccines, DNA-based CMV vaccines, dense body vaccines, etc.) are being developed [20]. If CMV vaccines become available, it is speculated that tests for detecting epitope-specific antibody and assays for measuring CMV-specific, T cell-mediated immunity may become indispensable tools for identifying CMV-seropositive pregnant women who require vaccination for preventing congenital infection caused by reinfection with a different strain.

However, CMV vaccines and established fetal or neonatal treatments against symptomatic congenital CMV infection are not yet available. In addition, a randomized, placebo-controlled, double-blind study had revealed that prophylaxis with CMV-specific hyperimmune globulin for the prevention of fetal CMV infection among pregnant women with primary CMV infection was ineffective [54]. Therefore, neither maternal nor neonatal universal screening for congenital CMV infection has been recommended worldwide.

However, recent studies have demonstrated that early intervention with antiviral drugs may improve neurological outcomes in infants with symptomatic congenital CMV infection. Therefore, prenatal detection of high-risk pregnancies with congenital infection of CMV is important. However, not only primary CMV infection during pregnancy, but also non-primary CMV infection can cause severe symptoms of congenital CMV infection. To detect all congenitally infected infants, universal neonatal screening based on CMV DNA PCR assays in newborn urine is necessary. However, we think that not only further studies for establishing fetal or neonatal treatments for symptomatic congenital CMV infection, but also cost–benefit analyses of universal neonatal CMV screening are required.

We show a graphic abstract as an example of a preemptive strategy for congenital CMV infection that was developed based on the results of our prospective cohort studies or clinical trials. Certainly, neonatal universal screening based on PCR assays for CMV DNA in newborns’ urine is preferable for detecting infants with congenital CMV infection without overlooking them. However, if universal CMV screening of newborn urine is not available, CMV DNA PCR tests in the newborn urine should be performed for newborns delivered by pregnant women who carry a high risk of congenital CMV infection. Namely, newborns at high risk for congenital CMV infection are detected by using the noninvasive biomarkers, including CMV serological tests, ultrasonography, PCR assay for maternal uterine cervical secretion, and clinical factors that may be associated with the occurrence of congenital CMV infection (e.g., the presence of threatened premature delivery, multiple pregnancies, maternal fever or flu-like symptoms, or preterm delivery). Newborns diagnosed with symptomatic congenital CMV infection by CMV DNA PCR tests in the urine and diagnostic tests for symptoms of congenital CMV infection receive antiviral treatments. This preemptive strategy may reduce the health and economic burden of the congenital CMV infection and disease.

## Figures and Tables

**Figure 1 ijms-19-03760-f001:**
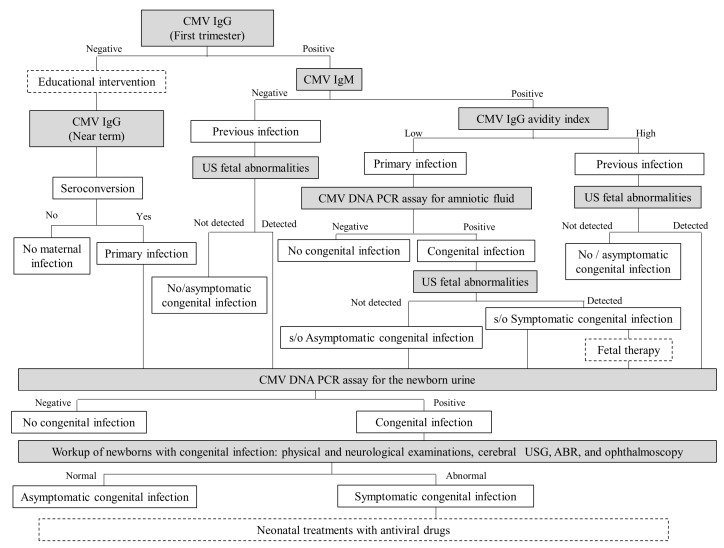
Flow algorithm for the diagnosis, prevention, and treatment of congenital CMV infection. Modalities for prediction or diagnosis of congenital CMV infection are indicated in gray. Interventions for preventing or treating congenital CMV infection are indicated in the dotted boxes. Abbreviations: CMV, cytomegalovirus; Ig, immunoglobulin; US, ultrasound; PCR, polymerase chain reaction; s/o, suspect of; No/asymptomatic, no or asymptomatic; USG, ultrasonography; ABR, auditory brainstem response.

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
