# Peer review of "Potential Biomarkers for Predicting Congenital Cytomegalovirus Infection"

_ijms, 2018, doi:10.3390/ijms19123760_

Round 1
Reviewer 1 Report
This review entitled, “Potential Biomarkers for Predicting Congenital Cytomegalovirus Infection” by K. Tanimura and H. Yamada is a well-written article that details regarding testing and screening of congenital CMV that predict (or could predict) congenital CMV infection. The review is well-organized, nicely outlines the current literature on the topic, and intercalates the authors’ opinions well. There are some spots throughout that could use references. Minor suggestions are provided below to improve readability and clarity:
1. The authors use the term “high risk pregnant women” and “low risk pregnant women”. While the reader could assume that these cohorts represent women how have not previously been infected vs. those who have, this is a big assumption (and it might not be what the authors mean). The authors should define/clarify what they mean by each phrase.
2. The authors use the phrase non-primary infection in several places. What does this mean? Does this mean reactivation? Or is this reinfection with a different strain?
3. Lines 12 and 120: “speculated to be the gold standard” – This is odd phrasing. It either is the gold standard (which means widely accepted) or it is not. “Speculated” should not precede “gold standard”.
4. Line 27: add “which” before “can be so severe…”
5. Line 38: Delete “the” that precedes “preemptive”
6. Lines 49-50: This sentence needs a reference.
7. Line 65: “recent and remote” – ‘Remote’ probably is not the right word here. Perhaps ‘previous’ is more appropriate.
8. Line 75: “Especially” is probably not the right word to start this sentence. Perhaps ‘specifically’ is more appropriate.
9. Line 96-97: “…majority of congenital CMV infection was caused by non-primary CMV infection” – While this goes along with point #2 above, this point should definitely be clarified. Is this due to reactivation or reinfection with a different strain?
10. Lines 113-114: This should be explained/clarified. AD169 and Towne are both lab-adapted strains (unless the “repaired” versions were used). Are gB and gH highly conserved between lab-adapted and clinical strains? If they’re highly conserved, how did their use serve as the basis for a method to distinguish different clinical isolates?
11. Line 150: add the word “a” between the words “between” and “group”
12. Line 151: Change “that without” to “those without”
13. Line 154: A reference is needed for the information in this paragraph.
14. Lines 175: consider changing “motivate clinicians” to “is the test result that motivates clinicians” (or something similar).
15. Line 195: Should MR be MRI?
16. Lines 220-221: These two hypotheses should read as such. The authors should consider adding phrases like “it is possible that” or “perhaps” after the two sentences that begin “First,….” and “Second,…”.
17. Lines 227-228: Reference is needed
18. Line 229: Reference is needed after “…false-positive results”
19. Line 234: Reference is needed after “…DNA in the urine.”
20. Line 250: Change “women were measure” to “women are measured”
21. Paragraph beginning on line 249: CMV seroprevalence world-wide is high. So where is the description regarding test results for a woman who is latently infected? That should be included/clarified.
22. Line 263: Delete ‘or’ before “into fetal…”
23. Line 264: Change “considered against fetuses” to “considered for fetuses”
24. Line 266: the phrase “as clinical trials” seems out of place.
25. Line 271: Change “concerning congenital CMV” to “indicative of congenital CMV”
26. Line 273: “worked up” is colloquial. Perhaps change to screened.
27. Figure 1: Is this standard of care at a specific institution or just the recommendation of the authors herein? This reviewer believes it’s the latter. If so, make that clear on Line 249. Also, the dark boxes are too dark, making the black text un-readable.
28. Line 290: Delete the comma after ‘infection’.
29. Line 300: insert a comma after ‘screening’.
30. Line 306: Add the word “who” before “carry a high risk”
31. Lines 310-311: Add parentheses around the “e.g.:” info.
32. Line 313: “work-up” – same comment as above. Perhaps here change to “tests”.
Reviewer 2 Report
Although this is an interesting and relevant review there are areas that need to be expanded to provide more general relevance to the field and specifically CMV infection detection vs outcome symptomatic vs asymptomatic cCMV/ primary infection/re-infection.
Why is there a greater risk of cCMV in primary infection during prgenancy compared to non-primary- this should be clarified. Despite some regions of the world being almost 100% CMV positive the underlying cCMV rate remain s high.........why?
Section 2,3 and 4. CMV specific antibody tests/ CMV IgG avidity/ Epitope specific antibody detection. These sections need to be re-worked to include a wider discussion on antibody response to specific target antigens. The gB glycoprotein is the only target antigen described and is misleading in that context. What are the neutralizing target antigens to HCMV. How do these contribute to protection. The viral pentameric or pentamer glycoprotein complex (PC) has emerged as an important immunological target to evaluate but there is no mention in the review. Indeed a deficiency in the response to the PC has been demonstrated as a risk factor for cCMV. The review mentions vaccines but the only vaccine to date to undergo indepth phase II studies is a gB subunit vaccine which only achieved 50% efficacy. Why?
Section 6 on T cell immunity should be xpanded to provide a more complete picture of what are the T cell targets and how does the specific T cell response to these target antigens factor into the protection against cCMV.
The statement made by the authors P4 lines 167-169 is confusing as it is unclear what is actually being measured- Is this primary CMV infection during pregnancy?
Section 7. Imaging examinations. This section should be redeveloped to provide specific stats associated with fetal abnormalities eg. microcephaly (approx. 4% cCMV).
Section 9. Should be expanded to discuss merits and results of neonatal screening in the context of newborn blood spots which potentially enables for faster evaluation of cCMV diagnosis. Guthrie cards are routinely used in the US for diagnosis of genetic diseases in newborns and so this enables easier screening of newborns regardless if urine screening is more sensitive.
Section 11. Although GCV therapy can potentially prevent further hearing loss in newborns with cCMV this has not been extensively evaluated and published. Perhaps a comaprison to the failure of HIG strategies to prevent cCMV should be included.
Although the focus of the review is on diagnostics, there is room in the conclusion section to provide some information on the development (or lack) of a vaccine against cCMV.
Round 2
Reviewer 2 Report
Point-by-point response to reviewers’ comments (For Reviewer #2)
Reviewer#2:
Comment 1: Why is there a greater risk of cCMV in primary infection during prgenancy compared to non-primary- this should be clarified. Despite some regions of the world being almost 100% CMV positive the underlying cCMV rate remain s high.........why?
Response: According to reviewer’s suggestion, in line 49-51, “Not only primary infection but also non-primary infection (reactivation or reinfection with a different CMV strain) can cause congenital infection, although the risk is decreased by 69% since the mother is seropositive [6].” was inserted.
R2.Revision comment. MODIFY to READ “decreased by UP TO 69%..”. Additionally, text should be modified to include reference to recent paper by Britt on the topic of non-primary cCMV.
Comment 2: Section 2,3 and 4. CMV specific antibody tests/ CMV IgG avidity/ Epitope specific antibody detection. These sections need to be re-worked to include a wider discussion on antibody response to specific target antigens. The gB glycoprotein is the only target antigen described and is misleading in that context. What are the neutralizing target antigens to HCMV. How do these contribute to protection. The viral pentameric or pentamer glycoprotein complex (PC) has emerged as an important immunological target to evaluate but there is no mention in the review. Indeed a deficiency in the response to the PC has been demonstrated as a risk factor for cCMV. The review mentions vaccines but the only vaccine to date to undergo indepth phase II studies is a gB subunit vaccine which only achieved 50% efficacy. Why?
Response: According to reviewer’s suggestion, in line 104-108, “Human CMV (HCMV) contains multiple surface-expressed glycoproteins that are critical to viral entry, including gB, the gM/gN complex, the gH/gL complex, and a pentameric gH/gL/UL128/UL130/UL131A complex. Recent studies indicate that the pentameric complex elicits significantly higher levels of neutralizing antibodies than gB or the gH/gL complex, therefore, the pentameric complex is a potential candidate for HCMV vaccines [15, 16].” was inserted, and 2 references “#15 Wen, Y., Monroe, J., Linton, C., Archer, J., Beard, C.W., Barnett, S.W., Palladino, G., Mason, P.W., Carfi, A., Lilja, A.E., Human cytomegalovirus gH/gL/UL128/UL130/UL131A complex elicits potently neutralizing antibodies in mice. Vaccine. 2014, 32(30), 3796-3804.” and “#16 Kobayashi, R., Abe, M., Oguri, K., Torikai, M., Nishimura, T., Mori, H., Koshizuka, T., Inoue, N., Analysis of relationships between polymorphisms in the genes encoding the pentameric complex and neutralization of clinical cytomegalovirus isolates. Vaccine. 2018, 36(40), 5983-5989.” were added.
R2. Revision comment. Anti-PC antibodies are not only more potent but have higher avidity to target antigen this should be clarified with appropriate refs. There is no discrimination in regard to the PC and why is is important for entry into specific cell types and why neutralizing abs have potential for specific prevention against cCMV. How does anti-PC reflect in avidity assays and ability to protect against cCMV. Text should be updated with more specific refs and clarification.
Comment 3: Section 6 on T cell immunity should be expanded to provide a more complete picture of what are the T cell targets and how does the specific T cell response to these target antigens factor into the protection against cCMV.
Response: According to reviewer’s suggestion, in line 172-174, “Many studies have demonstrated the critical role of CMV-specific T-cell-mediated immune responses in controlling viral replication and disease severity, however, the specific features of these responses that confer protection against fetal infection remain incompletely defined [24].” was inserted, and a reference “#24 La Rosa, C., Diamond, D.J., The immune response to human CMV. Future Virol. 2012, 7(3), 279–293.” was added.
R2 Revision comment. Still no mention of specific T cell target antigens! Surely response to specific T cell antigens is a biomarker of immune competence?
Comment 4: The statement made by the authors P4 lines 167-169 is confusing as it is unclear what is actually being measured- Is this primary CMV infection during pregnancy?
Response: CMV ELISPOT assay may be useful for predicting congenital CMV infection. According to reviewer’s suggestion, in line 181-182, “…, including reactivation or reinfection with a different strain).” was added. In line 186, “regardless of primary or non-primary infection [21].” was added.
Comment 5: Section 7. Imaging examinations. This section should be redeveloped to provide specific stats associated with fetal abnormalities eg. microcephaly (approx. 4% cCMV).
Response: According to reviewer’s suggestion, line 194-197 was changed to “Ultrasound fetal abnormalities, including ventriculomegaly (4.5%–11.6%), microcephaly (14.5%), intracranial calcification (0.6%–17.4%), FGR (1.9%–13%), pericardial effusion (7.2%), ascites (8.7%), hepatomegaly (4.3%), and intestinal high echodensities (4.5%–13%), are known to be predictive of symptomatic congenital CMV infection [11, 28-30].”, and references “#29 Picone, O., Vauloup-Fellous, C., Cordier, A.G., Guitton, S., Senat, M.V., Fuchs, F., Ayoubi, J.M., Keros, L.G., Benachi, A., A series of 238 cytomegalovirus primary infections during pregnancy: description and outcome. Prenat. Diagn. 2013, 33, 751-758.” and “#30 Guerra, B., Simonazzi, G., Puccetti, C., Puccetti, C., Lanari, M., Farina, A., Lazzarotto, T., Rizzo, N., Ultrasound prediction of symptomatic congenital cytomegalovirus infection. Am. J. Obstet. Gynecol. 23 198. 380.e1-7.” were added.
Comment 6: Section 9. Should be expanded to discuss merits and results of neonatal screening in the context of newborn blood spots which potentially enables for faster evaluation of cCMV diagnosis. Guthrie cards are routinely used in the US for diagnosis of genetic diseases in newborns and so this enables easier screening of newborns regardless if urine screening is more sensitive.
Response: According to reviewer’s suggestion, in line 251-256, “On the other hand, because dried blood spots are collected routinely for newborn metabolic screening from all infants born in the U.S., Japan, and so on, there has been considerable interest in neonatal universal screening for congenital CMV infection based on PCR assay in the newborn dried blood spots samples. However, a previous study demonstrated that real-time PCR analysis of dried blood spots had low sensitivity for correctly identifying newborns with congenital CMV infection [38].” was inserted, and a reference “#38 Boppana, S.B., Ross, S.A., Novak. Z., Shimamura. M., Tolan, R.W., Palmer, A.L., Ahmed, A., Michaels. M.G., Sánchez, P.J., Bernstein, D.I.; et al. Dried blood spot real-time polymerase chain reaction assays to screen newborns for congenital cytomegalovirus infection. JAMA. 2010, 303(14), 1375-1382. ” was added.
R2 revision comment. Discussion on the use of blood spots for screening appears biased against blood spot screening. Authors should cite specific refs that provide specific accurate detection of cCMV by blood spot and should provide specifics on limits of detection which is not provided in revision. If a side by side comparison from same patients urine vs blood then use specific ref to support their statement.
Comment 7: Section 11. Although GCV therapy can potentially prevent further hearing loss in newborns with cCMV this has not been extensively evaluated and published. Perhaps a comparison to the failure of HIG strategies to prevent cCMV should be included.
Response: According to reviewer’s suggestion, in line 320-323, “In addition, a randomized, placebo-controlled, double-blind study had revealed that prophylaxis with CMV-specific hyperimmune globulin for prevention of fetal CMV infection among pregnant women with primary CMV infection was ineffective [44].” was inserted, and a reference “#44 Revello, M.G., Lazzarotto, T., Guerra, B., Spinillo, A., Ferrazzi, E., Kustermann, A., Guaschino, S., Vergani, P., Todros, T., Frusca, T.; et al. A randomized trial of hyperimmune globulin to prevent congenital cytomegalovirus. N. Engl. J. Med. 2014, 370(14), 1316-1326.” was added.
Comment 8: Although the focus of the review is on diagnostics, there is room in the conclusion section to provide some information on the development (or lack) of a vaccine against cCMV.
Response: According to reviewer’s suggestion, in line 317-318, “Recently, it has been reported that the pentameric complex, i.e, gH/gL/UL128/UL130/UL131A complex, is a potential candidate for HCMV vaccines [15, 16].” was inserted.
R2 Revision comment. Probably better to additionaly include refs to recent reviews by Plotkin on cCMV vaccines. The PC is not the only Ag vaccine and additionally what vaccine approach- subunit, DNA, vector etc? Is a vaccine around the corner? What is the advantage of the use of biomarkers if a vaccine is to be shortly available? Specific updates will help put things in prespective.
Author Response
Reviewer#2:
Comment R2-1: MODIFY to READ “decreased by UP TO 69%.”. Additionally, text should be modified to include reference to recent paper by Britt on the topic of non-primary cCMV.
Response: According to reviewer’s suggestion, line 50-51 was changed to “…although preexisting maternal immunity against CMV is related to a 69% decrease in the risk of congenital infection [6].“, and in line 100-104, “Recently, many studies also demonstrated that the major of congenital CMV infection may be caused by non-primary CMV infection, and that the prevalence of symptomatic congenital CMV infection in infants born to mothers with non-primary infection is similar to that in infants born to pregnant women with primary infection [15].” was inserted. A reference “#15 Britt, W.J.; Maternal immunity and the natural history of congenital human cytomegalovirus infection. Viruses. 2018, 10(8), DOI 10.3390/v10080405.” was added.
Comment R2-2: Anti-PC antibodies are not only more potent but have higher avidity to target antigen this should be clarified with appropriate refs. There is no discrimination in regard to the PC and why is important for entry into specific cell types and why neutralizing abs have potential for specific prevention against cCMV. How does anti-PC reflect in avidity assays and ability to protect against cCMV. Text should be updated with more specific refs and clarification.
Response: According to reviewer’s suggestion, in line 110-112, “Especially, the pentameric complex is responsible for viral entry into endothelial, epithelial, monocytic cell, and dendritic cells, and the complex decides viral tropism for diverse cells [16, 17].” was inserted. Line 112-115 was changed to “…pentameric complex induces not only higher levels but also higher binding avidities of neutralizing antibodies than gB or the gH/gL, therefore, the pentameric complex is a potential candidate for human CMV vaccines [18-20].”, and a reference “#20 Schleiss, M.R.; Permar, S.R.; Plotkin, S.A.; Progress toward development of a vaccine against congenital cytomegalovirus infection. Clin. Vaccine. Immunol. 2017, 24(12), DOI 10.1128/CVI.00268-17.” was added.
In line 115-120, “In addition, it has been demonstrated that a delay in the production of maternal antibodies against the pentameric complex during primary infection is associated with mother-to-fetus CMV transmission [21], and that both antibodies against the pentameric complex and higher CMV IgG avidity index are correlated with decreased risk of congenital infection [22. Therefore, the pentameric complex is thought to be a major target for the development of vaccines to prevent fetal CMV infection.” was inserted, and references “ #21 Lilleri, D., Kabanova, A., Revello, M.G., Percivalle, E., Sarasini, A., Genini, E., Sallusto, F., Lanzavecchia, A., Corti, D., Gerna, G.; Fetal human cytomegalovirus transmission correlates with delayed maternal antibodies to gH/gL/pUL128-130-131 complex during primary infection. PLoS One, 2013, 8, e59863, DOI 10.1371/journal.pone.0059863.” and “#22 Kaneko, M., Ohhashi, M., Minematsu, T., Muraoka, J., Kusumoto, K., Sameshima, H.;. Immunoglobulin G avidity as a diagnostic tool to identify pregnant women at risk of congenital cytomegalovirus infection. J. Infect. Chemother. 2017, 23, 173–176.” were added.
Comment R2-3: Still no mention of specific T cell target antigens! Surely response to specific T cell antigens is a biomarker of immune competence?
Response: According to reviewer’s suggestion, in line 188-191, “Previous studies have demonstrated that pp65-specific CD8+ T cell and immediate-early antigen (IE1)-specific CD8+ T cells have a protective function against CMV viremia in transplant recipients [31, 32], and that pp65-specific CD4+ T cell may play a crucial role in the protection against mother-to-fetus CMV transmission [33].” was inserted, and references “#31 Gratama, J.W., van Esser, J.W., Lamers, C.H., Tournay, C., Löwenberg, B., Bolhuis, R.L., Cornelissen, J.J.; Tetramer-based quantification of cytomegalovirus (CMV)-specific CD8+ T lymphocytes in T-cell-depleted stem cell grafts and after transplantation may identify patients at risk for progressive CMV infection. Blood, 2001, 98, 1358–64.”, “#32 Bunde, T., Kirchner, A., Hoffmeister, B., Habedank, D., Hetzer, R., Cherepnev, G., Proesch, S., Reinke, P., Volk, H.D., Lehmkuhl, H., Kern, F.; Protection from cytomegalovirus after transplantation is correlated with immediate early 1-specific CD8 T cells. J. Exp. Med. 2005, 201, 1031–6.”, and “#33 Fornara, C., Cassaniti, I., Zavattoni, M., Furione, M., Adzasehoun, K.M., De Silvestri, A., Comolli, G., Baldanti, F.; Human cytomegalovirus-specific memory CD4_ T-cell response and its correlation with virus transmission to the fetus in pregnant women with primary infection. Clin. Infect. Dis. 2017, 65, 1659-1665.” were added. In line 204-206, “The author speculated that high cell-mediated immune responses may promote CMV transmission in primary infection, whereas the preexisting cell-mediated immunity in non-primary infection may exert protective effects against fetal infection [34].” was inserted. In line 210-211, “Thus, the results of assays for measuring CMV-specific T-cell-mediated immunity are sometimes difficult to interpret.” was inserted.
Comment R2-4: Discussion on the use of blood spots for screening appears biased against blood spot screening. Authors should cite specific refs that provide specific accurate detection of cCMV by blood spot and should provide specifics on limits of detection which is not provided in revision. If a side by side comparison from same patients urine vs blood then use specific ref to support their statement.
Response: According to reviewer’s suggestion, line 277-282 was changed to “However, in the largest prospective study, 20448 newborns were screened for congenital CMV infection by real-time PCR analysis of dried blood spots. This study demonstrated that the screening for congenital CMV infection based on the dried blood spots PCR tests yielded 28.3% sensitivity, and 99.9% specificity [47]. The same study group also reported that the screening for congenital infection using the PCR assays in the liquid-saliva yielded 100% sensitivity, and 99.9% specificity [48].”, and a reference “#48 Boppana, S.B., Ross, S.A., Shimamura, M., Palmer, A.L., Ahmed, A., Michaels, M.G., Sánchez, P.J., Bernstein, D.I., Tolan, R.W.Jr., Novak, Z.; et al. Saliva polymerase-chain reaction assay for cytomegalovirus screening in newborns. N. Engl. J. Med. 2011, 364(22), 2111–2118.” was added.
Comment R2-5: Probably better to additionaly include refs to recent reviews by Plotkin on cCMV vaccines. The PC is not the only Ag vaccine and additionally what vaccine approach- subunit, DNA, vector etc? Is a vaccine around the corner? What is the advantage of the use of biomarkers if a vaccine is to be shortly available? Specific updates will help put things in perspective.
Response: According to reviewer’s suggestion, in line 344-351, “Furthermore, the pentameric complex is thought to be a major target for the development of vaccines to prevent fetal CMV infection. On the other hand, albeit in a preclinical stage, new vaccines (e.g., enveloped virus-like particles vaccines, vectored CMV vaccines, DNA-based CMV vaccines, dense body vaccines, etc.) are being developed [20]. If CMV vaccines become available, it is speculated that tests for detecting epitope-specific antibody and assays for measuring CMV-specific T-cell-mediated immunity may become indispensable tools for identifying CMV-seropositive pregnant women who require vaccination for preventing congenital infection caused by reinfection with a different strain.” was inserted.
